# Cu_2_Se Nanoparticles Encapsulated by Nitrogen-Doped Carbon Nanofibers for Efficient Sodium Storage

**DOI:** 10.3390/nano10020302

**Published:** 2020-02-10

**Authors:** Le Hu, Chaoqun Shang, Eser Metin Akinoglu, Xin Wang, Guofu Zhou

**Affiliations:** 1National Center for International Research on Green Optoelectronics, South China Normal University, Guangzhou 510006, China; hule@m.scnu.edu.cn (L.H.);; 2International Academy of Optoelectronics at Zhaoqing, South China Normal University, Zhaoqing 526060, China; e.akinoglu@zq-scnu.org

**Keywords:** sodium ion batteries, Cu_2_Se-NC, carbon nanofibers, rate capability, cycling stability

## Abstract

Cu_2_Se with high theoretical capacity and good electronic conductivity have attracted particular attention as anode materials for sodium ion batteries (SIBs). However, during electrochemical reactions, the large volume change of Cu_2_Se results in poor rate performance and cycling stability. To solve this issue, nanosized-Cu_2_Se is encapsulated in 1D nitrogen-doped carbon nanofibers (Cu_2_Se-NC) so that the unique structure of 1D carbon fiber network ensures a high contact area between the electrolyte and Cu_2_Se with a short Na^+^ diffusion path and provides a protective matrix to accommodate the volume variation. The kinetic analysis and D_Na+_ calculation indicates that the dominant contribution to the capacity is surface pseudocapacitance with fast Na^+^ migration, which guarantees the favorable rate performance of Cu_2_Se-NC for SIBs.

## 1. Introduction

Sodium-ion batteries (SIBs) have garnered tremendous attention due to the low price and abundance of sodium resources, and are considered as one of the promising alternatives to commercial lithium-ion batteries [1,2,3]. However, the large ionic radius of Na^+^ causes huge volume variation during Na^+^ insertion/extraction, which results in sluggish reaction kinetics and undesirable electrochemical performance [4,5]. Thus, the exploration of suitable electrode materials is still a great challenge for effective and reversible accommodation of large Na^+^ for wide-scale practical application of SIBs [6,7].

Among various potential electrode material candidates for SIBs, metal selenides have attracted particular attention because of good electronic conductivity and high reversible capacity based on multielectron conversion reactions [8,9]. Lou et al. prepared Cu-doped CoSe_2_ microboxes with high reversible capacity of 492 mAh g^−1^, excellent rate capability and long cycling life [10]. Yang and co-workers synthesized a 3D trilayer (CNT/MoSe_2_/C) as freestanding electrodes with high areal capacity (4.0 mAh cm^−2^) and promising cycling stability [11]. Compared to MSe_2_ with high Se content, M_2_Se such as Cu_2_Se with an inactive Cu component has the advantages of high electronic conductivity, structural stability and low volume variation during cycling [12,13,14]. However, most of these alternatives suffer from large volume changes during the conversion reaction with unsatisfied electrochemical performances [15,16]. As an effective approach, designing nanostructured materials can increase the active sites and specific surface area, further improving the reaction kinetics and reversible specific capacity [17]. Nevertheless, nanomaterials tend to agglomerate during cycling, which still prevents the long-term cycle life of the batteries. To solve these issues, the common tactic is to synthesize metal selenide/carbon nanocomposites [18]. Nanosized active materials uniformly dispersed in carbon matrix can effectively inhibit aggregation of the nanoparticles and alleviate the volume expansion caused by Na^+^ insertion [19,20,21].

In this paper, Cu_2_Se nanoparticles are embedded in 1D nitrogen doped carbon nanofibers (Cu_2_Se-NC). The unique structure of 1D carbon fiber can shorten the Na^+^ diffusion path with a high contact area between Cu_2_Se and the electrolyte. Meanwhile, carbon conductive network provides a buffering matrix for the effective accommodation of volume variation and alleviates the aggregation of Cu_2_Se nanoparticles to maintain the structural integrity during repeated cycling. Based on the these favorable effects, Cu_2_Se-NC displays a promising rate capability and long-term cycling life for efficient sodium storage.

## 2. Experimental Section

### 2.1. Preparation of Cu_2_Se-NC Nanofibers

A solution containing 10 mL dimethylformamide (DMF, Aladdin, Shanghai, China), 0.4 g Cu(CH_3_COO)_2_·H_2_O (Aladdin), and 0.8 g polyacrylonitrile (PAN, *M*_W_ = 150,000, Aldrich, St. Louis, MO, USA) was vigorous stirred for about 12 h, which was then loaded into a 20-mL syringe with 19-G needle for electrospinning. The electrospinning process was applied voltage of 1 kV cm^−1^ with flow rate of 10 μL min^−1^. After electrospinning, the collected Cu(AC)_2_/PAN nanofibers were selenized with Se powder (amass ratio of 1:2) in tube furnace at 500 °C for 4 h (Ar/H_2_ atmosphere, heating rate: 2 °C min^−1^) to obtain Cu_2_Se-NC. The Cu_2_Se and nitrogen doped carbon nanofibers (NC) were acquired through the similar treatment procedure by using Cu(AC)_2_ with Se powder and PAN nanofibers, respectively.

### 2.2. Material Characterization

Field emission scanning electron microscope (FESEM, ZEISS Ultra 55, Carl Zeiss Inc., Oberkochen, Germany) and transmission electron microscope (TEM, JEM-2100F, JEOL Ltd., Tokyo, Japan) were used to observe morphology and structure. X-ray diffraction patterns (XRD, Bruker D8 Advance, Karlsruhe, Germany) and X-ray photoelectron spectroscopy (XPS, PHI 5600, Physical electronics, Chanhassen, MN, USA) were conducted to analyze the phase structure and chemical composition, respectively. Thermal gravimetric analysis (TGA, SDT Q600 TG-DTA, TA Instruments-Waters LLC, New Castle, DE, USA) was performed from room temperature to 600 °C (air, 10 °C min^−1^).

### 2.3. Electrochemical Measurements

The slurry composed of 80 wt% active material, 10 wt% Super P, 10 wt% polyvinylidene fluoride (PVDF) dispersed in appropriate *N*-methyl-2-pyrrolidone (NMP) was cast onto copper foil as electrode with active material loading of about 0.8~1.0 mg cm^−2^. The electrolyte was 1 M NaClO_4_, which dissolved in a solution of ethylene carbonate (EC) and dimethyl carbonate (DMC) with volume ratio of 1:1 as well as additive of 5 wt% fluoroethylene carbonate (FEC). The electrochemical performances tests were measured by a Neware battery testing system in a voltage window of 0.1~3.0 V. Cyclic voltammetry (CV) and electrochemical impedance spectroscopy (EIS) conducted on an electrochemical workstation (BioLogic VMP3, Bio-Logic, Seyssinet-Pariset, France).

## 3. Results and Discussion

As shown in Figure 1a, the morphology of the as-electrospun Cu(AC)_2_/PAN is composed of highly uniform and smooth nanofibers with diameter of ~200 nm. The selenized nanofibers maintained the 1D structure, where Cu_2_Se nanoparticles are uniformly distributed in the carbon nanofibers (Figure 1b,c). The appropriate void space between as-prepared Cu_2_Se-NC nanofibers is beneficial to the penetration of electrolyte. The low-resolution TEM images reveal that Cu_2_Se nanoparticles were homogenously embedded in the carbon nanofibers in Figure 1d, and further confirm the structural feature of Cu_2_Se-NC, agreeing well with the SEM results. A high-resolution TEM (HRTEM) image reveals the crystal phase of Cu_2_Se possess good crystallinity, and the lattice spacing of 0.204 nm is ascribed to the (220) crystal plane of Cu_2_Se (Figure 1e). Additionally, the selected area electron diffraction (SAED) of Cu_2_Se-NC indicates the formation of Cu_2_Se with polycrystallinity (Figure 1f). SEM image and corresponding EDX mapping results shown in Figure 1g,h reveal the uniform distribution of Cu, Se, N, and C throughout the nanofibers. However, bulk Cu_2_Se is composed of irregular nanoparticles in the absence of carbon nanofiber support (Appendix A).

The X-ray diffraction (XRD) pattern of as-prepared Cu_2_Se-NC is displayed in Figure 2a. The major diffraction peaks at 26.8, 31.0, 44.5, 52.6, 64.7 and 81.9° are ascribed to (111), (200), (220), (311), (400) and (422) lattice planes of Cu_2_Se (JCPDS No. 65-2982), which proves successfully synthesis of Cu_2_Se phase. In addition, a sharp and strong (220) peak indicates significantly crystallinity and in good agreement with the HRTEM and SAED characterization for Cu_2_Se–NC. As shown in the thermogravimetric analysis (TGA, Figure 2b), the weight change could be divided to several steps: (i) the temperature between 30 to 120 °C is likely due to the evaporation of water remaining; (ii) the 49.4% weight loss from 340 to 410 °C is attributed to the carbon combustion; (iii) the range of 410~550 °C could be estimated from the oxidation of Cu_2_Se into CuO, whereby the content of 29% is Cu_2_Se for Cu_2_Se–NC composite [22]. The elemental composition and valence state of Cu_2_Se–NC were investigated by XPS analysis (Appendix A). In the high-resolution Cu 2p spectrum (Figure 2c), the peaks at 952.4 and 932.6 eV are attributed to Cu^+^ in Cu_2_Se, and the peak at 954.4 and 934.6 eV could be assigned to the presence of Cu^2+^ in surface oxidized CuO [15,23]. For the XPS spectrum of Se 3d (Figure 2d), two major peaks located at 55.5 eV (Se 3d_3/2_) and 54.2 eV (Se 3d_5/2_) are ascribed to Cu-Se bonds [24]. The O 1s spectra (Appendix A) shows the peaks with binding energies of 532.4, 531.4 and 530.6 eV can be associated with absorbed water, C–O bond and Cu–O bond for Cu_2_Se–NC, respectively [25]. The N 1s spectrum (Appendix A) is deconvoluted into representative peaks centered at 401.5, 400.0, 398.9 and 398.3 eV, attributing to oxidized-N, graphitic-N, pyrrole-N and pyridine-N, further confirming the successful nitrogen doping into Cu_2_Se–NC composite [26]. The C 1 s region (Appendix A) is assembled in four components for the C–O (289.2 eV), C–N (286.3 eV), sp^3^ C–C (285.2 eV) and sp^2^ C=C bonds (284.6 eV) [27]. The existence of a carbon framework can improve the electronic conductivity and inhibit effectively volume expansion, which is favorable for fast electron transfer and structural stability during repeated cycling.

Figure 3a displays the CV curves of the Cu_2_Se-NC. At around 0.21 and 0.85 V of reduction process in the initial CV curve, there are obviously irreversible peaks, corresponding to the irreversible reduction of electrolyte and the formation of solid electrolyte interphase (SEI) film [28]. In the subsequent cycles, the broad reduction peak at ~0.7 V might be assigned to the formation of Na_2_Se, and the two oxidation peaks at 0.45 and 1.55 V are assigned to the extraction of Na^+^ from Na_2_Se. The CV curves are well overlapped from the second cycle, demonstrating excellent electrochemical reversibility of the Cu_2_Se–NC electrode. Figure 3b shows the charge–discharge profiles of the initial four cycles of Cu_2_Se–NC electrode at a current density of 0.1 A g^−1^. The Cu_2_Se–NC electrode yields a first discharge/charge specific capacity of 412/192 mAh g^−1^, which was higher than those of Cu_2_Se (Appendix A, 322/189 mAh g^−1^) and NC (Appendix A, 269/73 mAh g^−1^). The initial Coulombic efficiency of Cu_2_Se–NC electrode is 46.6%, which is consistent with the CV test results [29]. The cycling performance of the Cu_2_Se–NC, Cu_2_Se and NC anode for SIBs at a current density of 0.1 A g^−1^ is shown in Figure 3c. Cu_2_Se–NC maintains a reversible capacity of 172 mAh g^−1^ after 200 cycles. However, the capacity of Cu_2_Se is sharply decreased to 10 mAh g^−1^ from 189 mAh g^−1^ after 10 cycles, suggesting 1D carbon fiber network provides more stable structure networks for accommodation of volume expansion to ensure cycling stability. The specific capacity of NC is only 70 mAh g^−1^ due to the sluggish reaction kinetics of NC for SIBs.

The rate performance of the Cu_2_Se–NC, Cu_2_Se and NC anode is illustrated in Figure 3d. The Cu_2_Se–NC electrode exhibits a high specific capacity of 193, 176, 148, 126, and 99 mAh g^−1^ at the current densities of 0.1, 0.2, 0.5, 1, and 2 A g^−1^. The specific capacity of 188 mAh g^−1^ can be obtained, when current density is returned to 0.1 A g^−1^, revealing the outstanding reversibility of the Cu_2_Se-NC. While the capacity of Cu_2_Se and NC decreases to only 3 and 24 mAh g^−1^ at 2 A g^−1^, indicating a significant enhancement of the electrochemical performance resulting from nanosized-Cu_2_Se embedded 1D nitrogen doped carbon nanofibers. Moreover, as depicted in Figure 3e, the Cu_2_Se–NC also exhibits favorable long-term cycling performance. After activation for five cycles, the Cu_2_Se–NC anode shows capacity retention of 99% (101 mAh g^−1^) after 2000 cycles, further implying the strong stability of the 1D nanofiber structure.

The kinetic analysis was investigated to further explain the excellent rate capability of Cu_2_Se–NC. Figure 4a depicts the CV curves at various sweep rates (from 0.2 to 1.0 mV s^−1^). The relationship between the peak current (*i*.e., mA) and sweep rate (*v*, mV s^−1^) and obeys the equation of *i = av^b^* that *b* value of 0.5 or 1.0 implies diffusion-controlled or capacitive-controlled processes, respectively [30,31]. The *b* value is identified by the slopes of log (*i*) vs. log (*v*) in Figure 4b. The Cu_2_Se–NC possesses high *b* values for both cathodic (0.86, 0.84) and anodic (0.74, 0.78) peaks, indicating the surface capacitive-controlled processes. This is responsible for the fast reaction kinetics and ensures considerable rate performance. To qualify the predominated storage capacity contribution, the capacity can be divided into capacitive (*k*_1_) and diffusion (*k*_2_) according to *i = k_1_υ^1/2^ + k_2_υ* (*i/υ^1/2^ = k_1_ + k_2_υ^1/2^*) [32,33]. As shown in Figure 4c, the capacitive contributions of Cu_2_Se–NC at 0.6 mV s^−1^ are calculated as 64.9%, which indicates that Cu_2_Se–NC possesses fast Na^+^ intercalation/deintercalation. Furthermore, Figure 4d displays the surface pseudocapacitive contribution in the Cu_2_Se–NC increases to 72.6% as the sweep rate elevates to 1.0 mV s^−1^. The high ratio of surface capacitive contribution is beneficial to the rate performance, which agrees with the favorable rate capability of Cu_2_Se–NC (Figure 3d).

The EIS curves and equivalent electrical circuit of Cu_2_Se–NC and Cu_2_Se are shown in Figure 5 and inset. Before cycling, the charge-transfer resistance (*R_ct_*) of Cu_2_Se–NC and Cu_2_Se is 1225 and 242 Ω, which decreases to 130 Ω and increases 466 Ω after 100 cycles, respectively. The decrease in resistance of Cu_2_Se-NC electrode is due to the activation process during repeated cycling, resulting in faster electron transfer for SIBs. In contrast, the *R_ct_* of the Cu_2_Se increased significantly *R_ct_* after cycles, which might be ascribed to cracking and pulverization of Cu_2_Se caused by volume expansion. The diffusivity coefficient of sodium ions (*D_Na+_*) is obtained by EIS measurements based on *D = R^2^T^2^/*(*2n^4^A^2^F^4^C^2^b^2^*) and *Z′ = R_e_ + R_ct_ + bω^−1/2^* equations [34]. Here, *ω* is the low range angular frequency of EIS, and *b* is the Warburg factor that is the slope of the line *Z′ ~ ω^−1/2^*. Obviously, the slope *b* of Cu_2_Se–NC is much lower than that of Cu_2_Se after cycled, and the sodium-ion diffusion coefficient of Cu_2_Se–NC and Cu_2_Se are calculated as ~3.4 × 10^−21^ and 5.4 × 10^−22^ cm^2^ s^−1^, respectively. The higher *D_Na+_* of Cu_2_Se–NC electrode indicates fast diffusion kinetics for sodium storage.

To further prove the structural stability of the electrode, we investigate the morphologies of the Cu_2_Se–NC and Cu_2_Se electrode before and after 100 cycles. As illustrated in Figure 6a,b, the cycled Cu_2_Se–NC electrode is flat with regular nanofibers, which is similar to their original morphology (Appendix A). Meanwhile, the EDS mapping of the cycled Cu_2_Se–NC electrode in Appendix A shows uniform distribution of Cu, Se, N, and C in the nanofiber, demonstrating superior structural integrity during long-term cycling. In contrast, Figure 6c and d shows that the structure of cycled Cu_2_Se has collapsed into dense and massive bulk, which is in accordance with the rapid capacity decay. These results represent that 1D carbon fiber networks can maintain the superior structural integrity and strong tolerance of volume changes. Ex situ XRD analysis is performed to identify the Na-storage mechanism (Appendix A). The Cu_2_Se (220) peak could be detected in the pristine Cu_2_Se-NC electrode. When it is discharged to 0.1 V, two new peaks at 43.3° and 44.1° ascribe to (111) plane of Cu (JCPDS No. 04-0836) and (042) plane of Na_2_Se (JCPDS No. 47-1699), which indicates the insertion of Na^+^ into Cu_2_Se and the formation of Cu and Na_2_Se. After recharging to 3 V, the XRD pattern displays similar signals for Cu peak, which might be attributed to the irreversible reactions. Moreover, the Cu_2_Se (220) peak could not restore to the pristine state that could be due to crystalline Cu_2_Se to amorphous state, providing more active sites and fast reaction kinetics.

## 4. Conclusions

In summary, Cu_2_Se–NC was successfully fabricated and evaluated as anode materials for sodium storage. Corresponding CV kinetic analysis and D_Na+_ calculation indicates the Cu_2_Se–NC possesses dominant surface pseudocapacity contribution and fast Na^+^ migration, which ensures favorable rate performance for SIBs. Moreover, the carbon fibers networks afford a conductive and protective layer for the effective accommodation of volume changes to ensure their superior long cycle life. Thus, the as-prepared Cu_2_Se–NC electrode displays enhanced sodium storage properties with the high reversible capacity of 172 mAh g^−1^ (0.1 A g^−1^), considerable rate capability of 99 mAh g^−1^ (2 A g^−1^) and ultra-long cycling stability (99% capacity retention over 2000 cycles).

## Figures and Tables

**Figure 1 nanomaterials-10-00302-f001:**
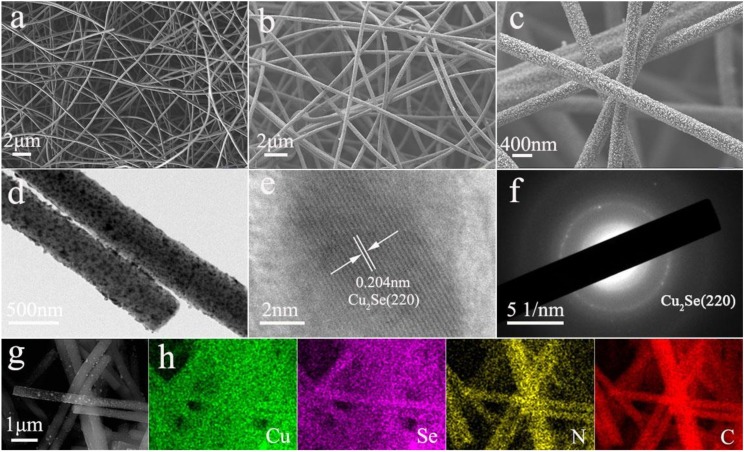
(**a**) SEM image of Cu(AC)_2_/PAN electrospun nanofibers; (**b**,**c**) SEM and (**d**) TEM images of Cu_2_Se-NC nanofibers after selenization treatment; (**e**) HRTEM image of Cu_2_Se-NC nanofibers. (**f**) Selected-area electron diffraction patterns of Cu_2_Se-NC nanofibers. (**g**) SEM image and (**h**) corresponding elemental mappings of Cu, Se, N, and C of Cu_2_Se-NC.

**Figure 2 nanomaterials-10-00302-f002:**
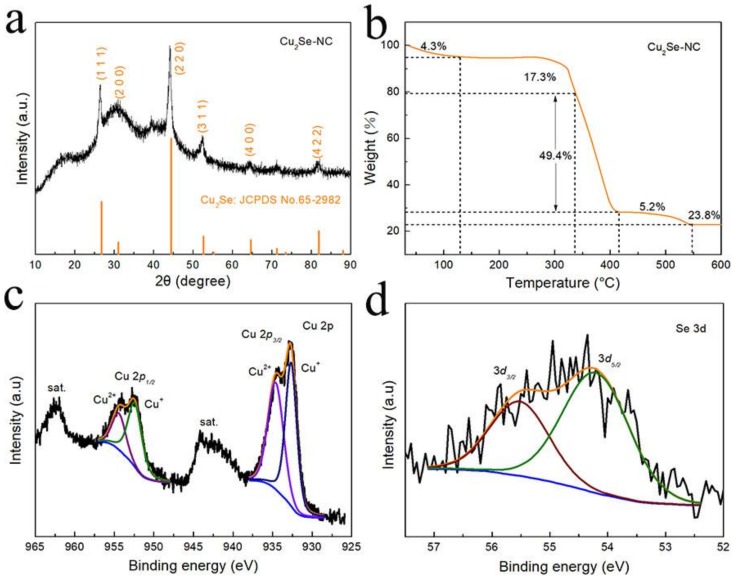
(**a**) XRD pattern of Cu_2_Se-NC; (**b**) TGA spectra of Cu_2_Se-NC; XPS spectra of (**c**) Cu 2p; (**d**) Se 3d of the Cu_2_Se-NC, respectively.

**Figure 3 nanomaterials-10-00302-f003:**
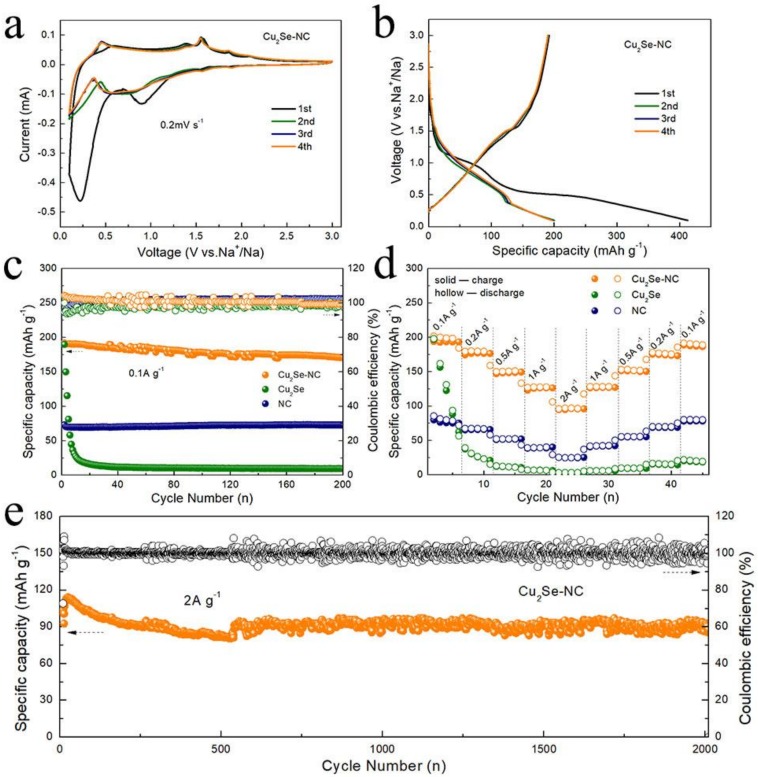
(**a**) Cyclic voltammograms of the Cu_2_Se–NC for the initial four cycles; (**b**) Charge/discharge curves of the Cu_2_Se–NC; (**c**) Cycling performance and of the Cu_2_Se–NC and Cu_2_Se and NC at 0.1 A g^−1^; (**d**) Rate capability of the Cu_2_Se–NC at various current densities; (**e**) Long cycling performance of the Cu_2_Se–NC at 2 A g^−1^.

**Figure 4 nanomaterials-10-00302-f004:**
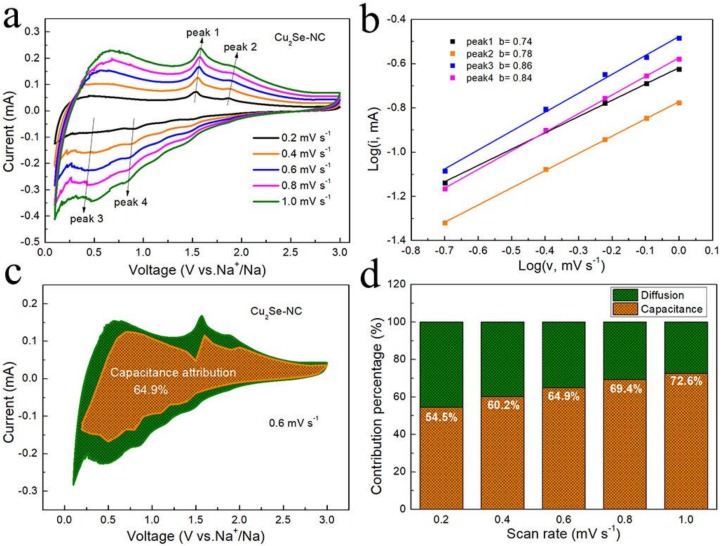
(**a**) CV curves at various scan rates of the Cu_2_Se-NC (0.2, 0.4, 0.6, 0.8 and 1.0 mV s^−1^); (**b**) log(*i*)-log(*v*) plots of the Cu_2_Se-NC; (**c**) capacitive contribution and diffusion contribution at 0.6 mV s^−1^; (**d**) capacitive and diffusion contribution versus scan rate of Cu_2_Se-NC.

**Figure 5 nanomaterials-10-00302-f005:**
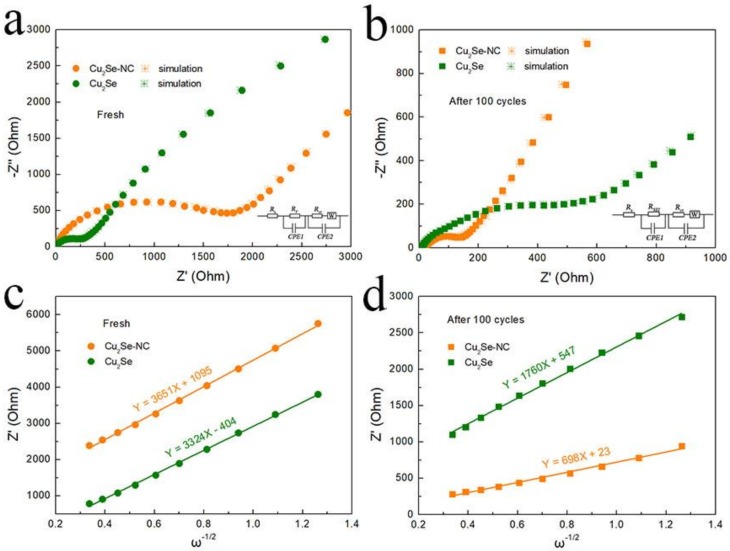
EIS data of Cu_2_Se–NC and Cu_2_Se electrodes at a current density of 0.1 A g^−1^ (**a**) before cycling, (**b**) after 100 cycles; the relationship plot between *Z′* and *ω^−1/2^* in the low-frequency region of (**c**) before cycling, (**d**) after 100 cycles.

**Figure 6 nanomaterials-10-00302-f006:**
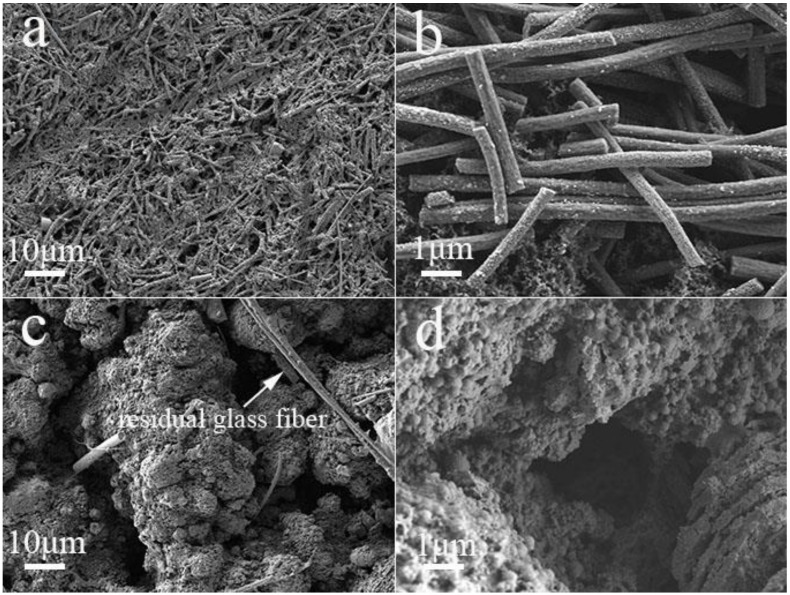
SEM images of (**a**,**b**) Cu_2_Se-NC and (**c**,**d**) Cu_2_Se electrode after 100 cycles.

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
