# Peer review of "Cu2Se Nanoparticles Encapsulated by Nitrogen-Doped Carbon Nanofibers for Efficient Sodium Storage"

_nanomaterials, 2020, doi:10.3390/nano10020302_

Round 1

Reviewer 1 Report

This paper is interesting to those who work in the field of Na-ion batteries.

I really liked the manuscript and also enjoyed the reading, because it is very well written and concise. Therefore I believe that there is enough merit in publishing as it stands.

With Best Regards,

The Rewiewer.

Author Response

Thank you for your high evaluation.

Reviewer 2 Report

The manuscript is in good form prepared and can be published as it is.

See my comment in attached file.

Author Response

Thank you so much for your high evaluation to our manuscript.

Reviewer 3 Report

The manuscript reports an interesting study, where the copper(I) selenide electrode material for Na-ion batteries was stabilized via encapsulation into N-doped carbon, obtained by reductive thermolysis of electrospun PAN-microfibers. The results are encouraging and the design of the material is promising. However, some points remain unclear:

1) In the figure 6C some microfibrils are visible. This material was produced without encapsulation - what is the nature and origin of fibrils?

2) It is important to compare in-depth the materials before and after electrochemical cycling for one and the same kind of material. The comparison of images in Fig. 6b and 1c demonstrates very considerable transformation. In 1c there are lighter dots, corresponding apparently to Cu2Se visible. In the 6c there are already relatively large raspberry constructions with lighter color. Is this an extensive diffusion of Cu2Se to the surface of the fibers? Is it correct that it does not effect so much the electrochemical performance under unvestigated conditions. For material in 6b it is necessary to provide EDS-mapping and XPS to get insight in not-so-small chemical transformaton.

Author Response

Reviewer#3:

The manuscript reports an interesting study, where the copper(I) selenide electrode material for Na-ion batteries was stabilized via encapsulation into N-doped carbon, obtained by reductive thermolysis of electrospun PAN-microfibers. The results are encouraging and the design of the material is promising. However, some points remain unclear:

Comment 1: In the figure 6C some microfibrils are visible. This material was produced without encapsulation - what is the nature and origin of fibrils?

Response: Thank you for your suggestion. Figure 6c depicts the SEM image of the cycled Cu2Se. The fiber is the residual glass fiber during battery disassembly, as marked in Figure 6c.

Comment 2: It is important to compare in-depth the materials before and after electrochemical cycling for one and the same kind of material. The comparison of images in Fig. 6b and 1c demonstrates very considerable transformation. In 1c there are lighter dots, corresponding apparently to Cu2Se visible. In the 6c there are already relatively large raspberry constructions with lighter color. Is this an extensive diffusion of Cu2Se to the surface of the fibers? Is it correct that it does not effect so much the electrochemical performance under unvestigated conditions. For material in 6b it is necessary to provide EDS-mapping and XPS to get insight in not-so-small chemical transformaton.

Response: Thank you for your suggestive comment. The initial Cu2Se-NC is that Cu2Se nanoparticles are uniformly dispersed in the carbon fiber (Figure 1c), corresponding to the pristine Cu2Se-NC electrode (Figure S5a and b). After the initial charge-discharge, Cu2Se-NC experienced some irreversible reaction and the formation of SEI film, as shown by the CV test (Figure 3a) and the first charge/discharge curves (Figure 3b). Besides, a slight volume expansion occurred during repeated cycles. Therefore, it is a normal phenomenon that the morphology of Cu2Se-NC has changed after 100 cycles (Figure 6a and b), and this is not an extensive diffusion of Cu2Se to the surface of the fibers.

The irreversible reaction and the formation of SEI film in the first cycle result in structural change of the cycled Cu2Se-NC, which may not be explained clearly by XPS and EDS for surface detection. The SEM comparison of Cu2Se-NC and Cu2Se before and after cycles is to clarify that 1D carbon fiber networks can maintain the superior structural integrity and strong tolerance of volume changes during sodium insertion/detraction, which further guarantees the excellent long-term cycling stability.

Round 2

Reviewer 3 Report

I do not feel that the authors responded to the original recommendation. The appearance of the fibers in 1c and 6b is really different with lighter raspberry constructions at 1 um in scale in 6b being much bigger than the dots in 1c at 400 nm in scale. At least EDS mapping is necessary to confirm the absence of extensive diffusion of the selenide component.

Author Response

Reviewer#3:

Comment 1: I do not feel that the authors responded to the original recommendation. The appearance of the fibers in 1c and 6b is really different with lighter raspberry constructions at 1 um in scale in 6b being much bigger than the dots in 1c at 400 nm in scale. At least EDS mapping is necessary to confirm the absence of extensive diffusion of the selenide component.

Response: Thank you for your suggestion. In the previous SEM image, there are many small balls on the surface of the cycled Cu2Se-NC, which might be derived from the precipitation of NaClO4 in electrolyte. To clarify this hypothesis, the cycled Cu2Se-NC was rinsed with DME for several times to remove the residual electrolyte and then for SEM test. As shown in Figure 6b, the cycled Cu2Se-NC shows similar morphology to the pristine Cu2Se-NC (Figure S5a and b), implying superior structural integrity. Furthermore, as illustrated in Figure S6, the EDS mapping of the cycled Cu2Se-NC with uniform distribution of Cu, Se, N, and C in the nanofiber, which confirm that Cu2Se do not diffuse to the surface of the carbon fiber during repeated cycling. In the revised manuscript, we have revised results and made corresponding discussion with red highlights as follows:

“Meanwhile, the EDS mapping of the cycled Cu2Se-NC electrode in Figure S6 shows uniform distribution of Cu, Se, N, and C in the nanofiber, demonstrating superior structural integrity during long-term cycling.”

Round 3

Reviewer 3 Report

The manuscript is revised and is fine for me now.